# Molecular Characterization and Pathogenicity of an Infectious cDNA Clone of Youcai Mosaic Virus on *Solanum nigrum*

**DOI:** 10.3390/ijms25031620

**Published:** 2024-01-28

**Authors:** Tianxiao Gu, Chenwei Feng, Yanhong Hua, Duxuan Liu, Haoyu Chen, Zhen He, Kai Xu, Kun Zhang

**Affiliations:** 1College of Plant Protection, Yangzhou University, Yangzhou 225009, China; mz120211383@stu.yzu.edu.cn (T.G.); mx120220848@stu.yzu.edu.cn (Y.H.); mz120231451@stu.yzu.edu.cn (D.L.); mx120230827@stu.yzu.edu.cn (H.C.); hezhen@yzu.edu.cn (Z.H.); 2Joint International Research Laboratory of Agriculture and Agri-Product Safety of Ministry of Education of China, Yangzhou University, East Wenhui Road No. 48, Yangzhou 225009, China; 3Jiangsu Key Laboratory for Microbes and Functional Genomics, Jiangsu Engineering and Technology Research Center for Microbiology, College of Life Sciences, Nanjing Normal University, Nanjing 210023, China; xukai@njnu.edu.cn

**Keywords:** youcai mosaic virus, infectious cDNA clone, *Solanum nigrum* L., phylogenetic trees

## Abstract

Virus infections cause devastative economic losses for various plant species, and early diagnosis and prevention are the most effective strategies to avoid the losses. Exploring virus genomic evolution and constructing virus infectious cDNA clones is essential to achieve a deeper understanding of the interaction between host plant and virus. Therefore, this work aims to guide people to better prevent, control, and utilize the youcai mosaic virus (YoMV). Here, the YoMV was found to infect the *Solanum nigrum* under natural conditions. Then, an infectious cDNA clone of YoMV was successfully constructed using triple-shuttling vector-based yeast recombination. Furthermore, we established phylogenetic trees based on the complete genomic sequences, the replicase gene, movement protein gene, and coat protein gene using the corresponding deposited sequences in NCBI. Simultaneously, the evolutionary relationship of the YoMV discovered on *S. nigrum* to others was determined and analyzed. Moreover, the constructed cDNA infectious clone of YoMV from *S. nigrum* could systematically infect the *Nicotiana benthamiana* and *S. nigrum* by *agrobacterium*-mediated infiltration. Our investigation supplied a reverse genetic tool for YoMV study, which will also contribute to in-depth study and profound understanding of the interaction between YoMV and host plant.

## 1. Introduction

*Solanum nigrum* L. (Nightshade), belonging to the genus *Solanum*, is an annual herb mainly found in temperate, subtropical, and tropical regions [1]. In some countries, *S. nigrum* is used as an edible food plant [2,3]. Generally, *S. nigrum* is widely used as a treatment for fever, swelling, ulcers, indigestion, and other ailments due to the multiple biological activities of its secondary metabolites. It is well known that plant viruses are widespread in nature, and the damage caused is often enormous. In natural conditions, *S. nigrum* is infected by a variety of plant viruses that have been reported previously, including alfalfa mosaic virus (AMV) [4], beet western yellows virus (BWYV) [5], cucumber mosaic virus (CMV) [6], cucurbit aphid-borne yellows virus (CABYV) [7], obuda pepper virus (ObPV) [8], pepino mosaic virus (PepMV) [9], pepper veinal mottle virus (PVMV) [10], plum pox virus (PPV) [11], potato spindle tuber viroid (PSTVd) [12], potato virus S (PVS) [13], potato virus Y (PVY) [14], tobacco mosaic virus (TMV) [15], tomato spotted wilt orthotospovirus (TSWV) [16], and tomato yellow leaf curl virus (TYLCV) [17]. In nature, whether other viruses also infect *S. nigrum* is an issue of concern. Clarification of this issue is good for proposing controlling strategies and will benefit deeper research of the interaction between *S. nigrum* and virus, and which will also have a positive impact on the development and utilization of *S. nigrum* resources.

Youcai mosaic virus (YoMV), which also known as oilseed rape mosaic virus (ORMV), belongs to subgroup III of the genus *Tobamovirus* [18]. YoMV is a positive-sense, single-stranded RNA virus with rod-shaped viral particles [19]. YoMV encodes four proteins, including the small replicase subunit (125 kDa), the large replicase subunit (182 kDa), the movement protein (MP), and the coat protein (CP). The 182 kDa protein replicase subunit has RNA-dependent RNA polymerase activity which is essential for viral genome RNA replication in vivo [20]. The 125 kDa protein is a replication-associated protein which also works as an RNA silencing suppressor (RSS) [21,22]. YoMV was first identified in China on oilseed rape and *Brassica napus* in 1958 [23]. YoMV infects plants belonging to the *Cruciferae*, *Liliaceae*, and *Brassicaceae* family, and is distributed worldwide [24,25,26]. YoMV infection causes devastating disease on rapeseed and can lead to yield losses of a great extent in some severe cases. Spread of YoMV mainly depends on mechanical transmission by the tiny wound on the cell surface. In addition, due to its wide host range, YoMV also poses a big threat to the growth and development of other traditional Chinese herbs, such as libosch (*Rehmannia glutinosa*), bell pepper (*Capsicum annuum*), fleabane (*Erigeron annuus*), and yam (*Dioscorea opposita*) [27,28,29,30]. Hence, for the safe production of these traditional Chinese herbs in areas where they serve as a pillar industry, construction of the *Agrobacterium*-infiltration-based infectious cDNA clone of YoMV and further exploring the underlying molecular interaction between virus and host are urgent. Understanding the origin and evolutionary relationships of YoMV is also essential for better diagnosis, controlling, and utilization of the relevant virus isolates.

In this study, we first found that YoMV infects *S. nigrum* in natural conditions. Then, the *Agrobacterium*-infiltration-based infectious cDNA clone of the YoMV Yangzhou isolate (YoMV-YZ) was constructed using in vivo recombination in yeast. The cDNA clone of YoMV was able to systemically infect the *Nicotiana benthamiana* and *S. nigrum* plants, and caused different symptoms. Furthermore, we constructed the phylogenetic tree based on the clone’s full-length genomic RNAs sequences of YoMV-YZ and other YoMV isolates deposited in NCBI. Simultaneously, the phylogenetic trees were based on different viral coding sequences. The results showed that the YoMV-YZ that was isolated from *S. nigrum* was most closely related to the YoMV isolated in Shanghai, China. Moreover, we found YoMV was widely distributed in *S. nigrum* that collected from different cities of Jiangsu Province. The investigation expanded our understanding of the molecular characteristics of YoMV. Furthermore, the infectious cDNA clone of YoMV provided a reverse genetic tool for further study of the interactions between YoMV and plants, which will contribute to a deeper investigation of the virus’s characteristics and pathogenicity, and also lay a fundamental basis for further research.

## 2. Results

### 2.1. S. nigrum Samples Exhibited Virus-Infection-Like Symptoms in Campus

On 24 September 2022, *Solanum nigrum* exhibited curling, mosaic, and shriveling symptoms in the campus of Yangzhou University in Jiangsu Province, China (Figure 1). Symptomatic samples (S1–S4) were collected, freeze-dried, and stored in a −80 °C refrigerator for further analyses.

### 2.2. Identification and Detection of YoMV in S. nigrum

To confirm whether these symptomatic *S. nigrum* were infected by plant viruses, we extracted total RNAs from these four symptomatic samples. After removing the other types of RNA, the siRNA was isolated and purified. Then, the siRNA libraries were constructed by EBNext Ultra II RNA Library Prep Kit (Cat. E7770L, NEB). Then, high-throughput siRNA sequencing was performed. A total of 4,814,450 reads were obtained. After removing sequences, such as linkers and low-quality reads, there were 3,267,897 valid reads. Most of the reads in the total reads or unique reads were distributed in the 21 (53.3%) to 24 nt (52.5%) length region (Figure 2A). After de novo assembly using velvet software and BLASTN analyses, 95 contigs were hit to different virus genomic RNAs with relative highest identity. The results showed that the youcai mosaic virus (YoMV) and tomato pseudo-curly top virus (TPCTV) existed in these symptomatic *S. nigrum* plants (Appendix A). Then, to verify the accuracy of the siRNA sequencing results, we designed specific pairs of primers to amplify the *CP* genes of YoMV (474 bp) and TPCTV (729 bp) (Appendix A). RT-PCR results showed that sample 1, sample 2, and sample 4 were infected by YoMV (Figure 1G), and sample 1, sample 2, and sample 3 were infected by TPCTV (Figure 1F). Due to the wide host range and severe economic losses caused by YoMV on agriculture, we chose the YoMV as material for further research. In sum, high-throughput siRNA sequencing and RT-PCR analyses showed that *S. nigrum* was infected by YoMV in the wild field.

To further investigate whether the *S. nigrum* was infected by YoMV, an additional eighteen *S. nigrum* samples (S5–S22) in the campus were randomly collected. YoMV infection was determined using enzyme-linked immunosorbent assay (ELISA) and RT-PCR. S1, S3, S4, S6, S8, S9, S11, S12, S15, S17, S18, S19, S21, and S22 showed as positive in ELISA analyses (Figure 2B). These samples showed a relatively higher record of OD_405_ than that of the healthy sample (Figure 2B), which was parallel to the positive control. In RT-PCR assay, using specific pairs of primers to the corresponding *CP* gene of YoMV and TPCTV, a total of 14 samples successfully appeared amplicons with the expected size that as the positive control of YoMV and TPCTV, which indicates these samples were YoMV-infected (Figure 2C). RT-PCR results were in line with the ELISA assay (Figure 2B,C). The results showed a 63.6% occurrence of YoMV on *S. nigrum* that was collected in the campus of Yangzhou University (Yangzhou, China).

To further determine the occurrence and distribution of YoMV in cities located in Jiangsu Province, an additional twenty-eight *S. nigrum* samples with virus-infection-like symptoms were randomly collected from different cities of Jiangsu Province, China. RT-PCR was performed to detect the YoMV using the specific primers (Appendix A); the results showed that the YoMV occurrence was 71.4% on *S. nigrum* in Jiangsu Province, and the highest occurrence of 80% was identified in Yancheng City (Appendix A). These results demonstrated that YoMV infection of *S. nigrum* were widely existing in Jiangsu Province, China.

What the origin is of YoMV-YZ appearing on *S. nigrum* plants has sparked new thinking. Has the virus undergone some changes while crossing hosts and geographic locations? What kind of differences exist between YoMV on *S. nigrum* plants and other previously reported YoMV isolates? Based on these issues, we aimed to obtain the complete genome of YoMV-YZ on *S. nigrum*, looking forward to better understanding the above questions of YoMV-YZ isolate.

### 2.3. Construction of the YoMV Full-Length cDNA Infectious Clones

To study the molecular characterizations of YoMV on *S. nigrum*, we attempted to construct the full-length cDNA infectious clone of YoMV-YZ isolate (Figure 3A). Firstly, we performed sequence alignment of the complete genome sequences of different YoMV strains deposited in NCBI. Fortunately, we found that all these YoMV genome sequences have relative conserved 5’ sequences, and only two different bases in the first 30 bp. Hence, we designed the degenerate forward primer of YoMV-YZ-F. Although the 3’ sequences of all isolates appear to be different, there are three types, and the degree of conservation of these sequences is clearly related to the geographical distribution: we further analyzed all the YoMV isolates within the Chinese mainland and found that these isolates have conserved 3’ sequences and only one base was different, so we designed the degenerate primer YoMV-YZ-R on this basis (Appendix A). Our primers are suitable for amplifying the complete genome sequences of different YoMV strains within the Chinese mainland (Appendix A). Due to the relative larger size of the complete genome of YoMV, it is difficult to direct amplification of the whole genome once. Therefore, we divided the genome sequence of YoMV into two fragments, including the 3440 bp frag1 and 2892 bp frag2 (Figure 3B). According to the high-throughput sequencing and BLASTN results, we designed two specific pairs of primers corresponding to frag1 and frag2. We used the cDNA of YoMV-infected *S. nigrum* samples as a template, and the target fragments were amplified. Reverse transcription was performed using the reverse primers YoMV-YZ-R1 and YoMV-YZ-R, generating the cDNA-R and cDNA-R1 products, respectively. Then, these cDNA-R and cDNA-R1 products were used as templates, and fragment 1 and fragment 2 were amplified using the pair of primers YoMV-YZ-F/YoMV-R1 and YoMV-YZ-F1/YoMV-YZ-R, respectively (Figure 3C). All the primer details are listed in the Appendix A.

Then, the purified two fragments and the linearized pCA4Y vector were co-transformed into the yeast strain *YPH500*, and the positive yeast colonies were screened via colony PCR. Due to their existing 25 bp or longer overlapping sequences between these three fragments (frag1, frag2, and linearized vector), the two full-length YoMV cDNA sequence was easy to obtain using a yeast recombination system [31] (Figure 3D). Positive pCA4Y-YoMV-YZ plasmid was obtained and transformed into *Agrobacterium* EHA105 strain. The infectivity of the constructed YoMV cDNA infectious clone was validated by *Agrobacterium*-mediated infiltration on *N. benthamiana* and *S. nigrum*. At 5 dpi, symptoms such as leaf shriveling and curling, together with slight necrosis, were observed in the upper leaves of the inoculated plants (Figure 3E, left panel). Western blot results showed that the constructed YoMV cDNA infectious clone could systemically infect *N. benthamiana* using the self-prepared specific anti-CP^YoMV^ antibodies (Figure 3F, upper panel). Using *CP* gene-specific primers, RT-PCR was performed and detection was the same as the results obtained from Western blot (Figure 3F, upper panel). All these results showed that *N. benthamiana* plants inoculated with the constructed YoMV cDNA infectious clone were systemically infected by YoMV, which indicates the construction of the YoMV cDNA infectious clone was successful.

To test whether the YoMV infectious cDNA clone infected the origin host that we first identified (*S. nigrum*), the *Agrobacterium* containing pCA4Y-YoMV-YZ plasmid was infiltrated into five-leaf stage *S. nigrum*. At 7 dpi, the symptoms, including leaf curling in systemic leaves and leaf mosaic, appeared on the upper leaves of infiltrated plants (Figure 3E, right panel). Western blot and RT-PCR assays showed that the *S. nigrum* plants were indeed infected by YoMV (Figure 3F,G). We performed 10 repetitions on ten individual plants for each YoMV infectious cDNA clone. As expected, all of the inoculated plants were systemically infected, with the results showing an infection rate of 100%. In conclusion, the YoMV infectious cDNA clones were successfully constructed, and they could systemically infect the model plant *N. benthamiana* and the native host *Solanum nigrum*.

### 2.4. Sequence Analysis

To further understand the evolution and recombination of YoMV in nature, we sequenced the three successfully constructed infectious cDNA clones of YoMV, and obtained three different complete genome sequences of YoMV-YZ isolates (OR261028, OR261029, and OR261030). All YoMV-related complete genome sequences deposited in the NCBI database were downloaded, and further alignment analyses were performed. First, all 22 full-length YoMV genome sequences, together with our sequences, were submitted for analysis, and a phylogenetic tree based on the genome sequences was constructed (Figure 4A). The results showed that our three YoMV-YZ isolates formed a self-contained cluster and were most closely related to the YoMV-SH isolate (AF254924), and most distantly related to the YoMV South Korea isolates (MW307290) (Figure 4A). The same results were obtained when using the genome organization and pairwise identity analysis (Figure 4B).

Furthermore, the coding sequences for viral replicase (RdRp), movement protein (MP), and coat protein (CP) that were deposited in NCBI were selected and then subjected to multipole sequences alignment and evolution analyses (Figure 4C,F). In terms of the RdRp coding sequences, YoMV-YZ isolate was highly consistent with the YoMV-SH isolate (AF254924) (Figure 4C,D), which is in line with the results of the genomic analysis. In terms of the MP and CP coding sequences, YoMV-YZ isolate was more closely related to YoMV-CQ isolates (MT130444) (Figure 4E,F and Appendix A). Based on these findings, we can conclude that the YoMV-YZ isolate shows a high degree of similarity to YoMV-SH isolate (AF254924). Compared with other isolates in the NCBI database, YoMV-YZ isolates were more likely to come from other Provinces in China. In addition, the evolutionary relationships between different YoMV strains isolated from different regions and different hosts were closely related to the spatial distance, which shows that the closer the spatial distribution, the stronger the evolutionary relationship, and *vice versa*.

## 3. Discussion

In this study, it was found that YoMV-YZ could infect *S. nigrum* in natural conditions, and it also confirmed that *S. nigrum* infected by YoMV-YZ was widely distributed in the cities of Jiangsu Province. All of this implies the urgency of proposing preventing and controlling strategies of YoMV on *S. nigrum* in future research. Our study firstly confirmed the infection of YoMV on *S. nigrum* in natural conditions, and also evidenced that infection of YoMV on *S. nigrum* was common worldwide (Figure 1 and Figure 2). The occurrence rate of YoMV on *S. nigrum* plants in Jiangsu Province was 71.4%, and was very common (Appendix A). This finding has changed our traditional understanding of the characteristics of YoMV. The geographical distribution of YoMV-infected *S. nigrum* was clearly mapped in Jiangsu Province (Appendix A). In addition, as a widely distributed weed in fields, *S. nigrum* has essential medicinal value [32,33,34]. In China, a very important strategy to improve the incomes of areas in poverty are the planting of *S. nigrum* for medical usage. However, our results implied that YoMV poses a significant threat to the safe production of *S. nigrum* in areas of poverty in China. *S. nigrum*, as a widespread plant, is a natural intermediate host for some other plant viruses that significantly threaten horticulture crops. Hence, it is most necessary to deeper understand the molecular characteristics of YoMV on *S. nigrum* from the perspective of host defense and viral counter defense, which is crucial to minimize the potential risk of a plant virus outbreak and pandemic.

Here, we successfully obtained the full-length genomic sequences of the YoMV-YZ isolates, and constructed the *Agrobacterium*-infiltration-based cDNA infectious clone of YoMV-YZ (Figure 3C). The constructed YoMV-YZ infectious clone was able to systemically infect both the model plant *N. benthamiana* and its natural host *S. nigrum* using *Agrobacterium*-mediated infiltration (Figure 3D). Systemic leaves of *N. benthamiana* and *S. nigrum* plants with *Agrobacterium*-mediated infiltrated plants displayed typical symptoms of YoMV infection (Figure 3D), and the molecular detections also confirmed the successful construction of the reverse genetic tool of YoMV-YZ (Figure 3E,F). We are looking forward to the wide application of this tool in exploring the interactive relationship between YoMV and hosts, including, but not limited to, elucidating the pathogenic mechanism of YoMV on tomato and oilseed rape. Although *Agrobacterium*-infiltration based YoMV cDNA infectious clones of other isolates have already been successfully established in recent years [35,36], it is still significant to construct the YoMV-YZ cDNA infectious clones occurring on *S. nigrum*, which will benefit for uncovering the molecular interaction mechanism between virus and plant by comparative study of different isolates in future studies. The constructed *Agrobacterium*-infiltration based YoMV-YZ infectious cDNA clone supplied a valuable tool and material in future study, which also laid the foundation for further deeper comparative studies in plant virology. The presented results also provided a variety of molecular diagnostic assays and sequence analyses of YoMV. The constructed infectious cDNA clones of YoMV-YZ could successfully infect its original host (*S. nigrum*), and the displayed symptoms are basically the same we observed in campus (Figure 1E and Figure 3D, right panel). In addition, we also noted that the leaf curling on *S. nigrum* we observed was mainly caused by TPCTV infection. With YoMV infection on *S. nigrum*, the symptoms were often milder, as the results observed in YoMV-YZ cDNA infectious clone infection on *S. nigrum* in the laboratory. Compared to the symptoms caused by YoMV or TPCTV infection alone (Figure 1C,D), the mixed infection of the two viruses simultaneously resulted in more promoted and severe symptoms of leaf curling, mottling, and coloration, which is also in agreement with the common understanding of mixed infection that promoted each virus’s infection. In natural conditions, mixed infections of the different plant viruses are very common on the same plant [37,38]. The antagonism or synergism of the two infected target viruses plays an important role in symptoms displayed on the same individual plant [39]. Irene et al. found that co-infection with tomato yellow leaf curl virus (TYLCV) and tomato chlorotic virus (ToCV) made symptoms more pronounced than that with a single virus infection [40]. Additionally, a significant co-infection symptom was observed in zucchini when a motility-deficient cucumber mosaic virus (CMV) was co-infected with zucchini yellow mosaic virus (ZYMV) [41]. However, further studies are needed on the clear pathogenic mechanisms and synergistic effects of TPCTV and YoMV.

Complete genomic sequences of YoMV-YZ isolate were compared with corresponding sequences of other YoMV isolates from different regions or hosts, and several phylogenetic trees were constructed (Figure 4). It was found that the YoMV-YZ isolate was located in a relatively autonomous cluster and displayed the highest sequence identity and closest relationship with the YoMV Shanghai isolate (AF254924) (Figure 4A). This may be attributed to the relatively close geographic proximity of the cities of Yangzhou and Shanghai. Additionally, Shanghai city is a prominent commercial center for frequent importing and exporting trades worldwide, which also facilitates plant viruses’ rapid transmission and spread in natural conditions. Phylogenetic trees suggested less variation among the same viral species from the same host, and same region, which is consistent with the evolutionary pattern of most plant viruses [42]. There is a clear positive correlation between the presence of YoMV in different geographic regions and their spatial proximity (Figure 4 and Appendix A). Interestingly, the YoMV-YZ isolate is most closely related to the YoMV-SH strain (AF254924), but the movement protein and coat protein parts of YoMV-YZ isolate are most closely related to the Chongqing strain and have higher sequence identity, which may be due to natural recombination of the virus for adapting to different environments and hosts. It has also been demonstrated that a change in amino acids of some of the movement protein and coat proteins of YoMV can lead to different symptoms in infected plants [36]. Comparing the sequences of the *MP*/*CP* part with different YoMV strains around the world, the YoMV-YZ strain is more closely related to other strains in China and was most distantly related to strains in South Korea (MW307290). The sequences of these two parts of YoMV-YZ still show a positive correlation with spatial distances around the world. Although the biological significance of these mutations or recombination in YoMV-YZ remains unclear, mutations and recombination in RNA viruses bear significant importance for generating genetic variation, adapting to new hosts, evading host antiviral immune responses, and emerging as new infectious agents [43]. However, further research is necessary to gain more profound insights.

## 4. Materials and Methods

### 4.1. Plant Cultivation

Test plants (*Nicotiana benthamiana* and *Solanum nigrum*) were grown in a chamber at 25 °C with a 16/8 h photoperiod (16 h of full levels of light and 8 h of dark treatment).

### 4.2. RNA Extraction and RT-PCR Detection

Total RNAs were extracted using TRIzol Reagent (Invitrogen, Waltham, MA, USA). The reverse transcription was performed to generate the first strand cDNA with commercial kit (RR037A, TakaRa Bio, Dalian, China). All operations were performed strictly according to the manufacturer’s instructions. Specific pairs of primers (Appendix A) were designed for first strand cDNA synthesis and RT-PCR detection based on the results of high-throughput siRNA sequencing. All the primers were synthesized by General Biosystems (Anhui) Co., Chuzhou, China.

### 4.3. High-Throughput siRNA Sequencing

*S. nigrum* samples were collected and mixed in equal amounts. Total RNAs were extracted, the small interfering RNAs (siRNA) were isolated and purified. Then, the siRNA libraries were prepared as described previously [44]. The Illumina Hiseq 2000 platform was used to perform the siRNA sequencing by Shanghai Sangon Bioengineering Co, Shanghai, China. The siRNA sequencing data were trimmed, and assembled using Virus Detect Platform [45]. BLASTN analyses with the siRNA sequencing data on the NCBI database were performed, and the results with viral matches are shown in the Appendix A.

### 4.4. Enzyme-Linked Immunosorbent Assay (ELISA) Detection

Protein extraction procedure is listed as follows: Leaves (0.1 g) of fresh *S. nigrum* were frozen in liquid nitrogen and ground with a grinder (Shanghai Jingxin Industrial Development Co., LTD., Shanghai, China), and then the total proteins were extracted as described [46]. After centrifugation, the supernatant of *S. nigrum* (100 μL) was loaded into 96-well polystyrene plates (Costar, Corning Inc., Corning, NY, USA). Then, the plate was placed in 37 °C incubator for 2 h. After washing and blocking, the self-prepared anti-CP^YoMV^ polyclonal antibodies were used to detect the YoMV. Each sample was replicated three times and averaged. The chromogenic reaction and termination reaction methods used in the analysis were consistent with the methods previously described [47]. The results were recorded using the ELISA reader (PowerWave XS2, BioTek Instruments, Santa Clara, CA, USA) at OD_405_. Data obtained from ELISA reader were treated by Excel software (Version 16.76.1), and the average reading was calculated. If the absorbance ratio of OD_405_ of the tested sample to the negative quality control was greater than 2-fold, we judged it as positive.

### 4.5. Construction of the cDNA Infectious Clone of YoMV-YZ

For constructing cDNA infectious clone of YoMV-YZ isolates, two overlapping fragments of YoMV, including Frag.1 (3440 bp) and Frag.2 (2892 bp), were amplified using the specific pairs of primers from prepared cDNA from YoMV-infected S. nigrum. Fragments 1 and 2 products were purified using Agarose Gel DNA Extraction Kit (Cat. 11696505001, Roche, Shanghai, China). Purified Frag.1, Frag.2, and *Stu* I/*Bam*H I-digested pCA4Y vector were mixed in vitro in a 2:2:1 molar ratio, and then transformed into yeast YPH500 strain as described previously [31]. Subsequently, a positive cDNA clone of full-length YoMV genomic RNA was obtained [31]. We provide a list of all pairs of primers used in the Appendix A.

### 4.6. Agrobacterium-Infiltration-Based Inoculation of the YoMV-YZ

The constructed pCA4Y-YoMV-YZ plasmid was transformed into *Agrobacterium* tumefaciens strain EHA105 using the freeze-thaw method [48]. Positive clones were selected in a kanamycin and rifampicin antibiotics LB plate, and further validated via PCR analyses. *Agrobacterium* tumefaciens transformed with pCA4Y empty plasmid was used as a negative control. *Agrobacterium*-infiltration based inoculation was performed as described [49] Briefly, the OD_600_ of the agrobacterium cell suspension was adjusted to 0.3 and the samples were incubated at 28 °C for two hours. A total of 5–6 leaf stages of *N. benthamiana* and *S. nigrum* were infiltrated.

### 4.7. Western Blot

The upper symptomatic leaves of *N. benthamiana* and *S. nigrum* were collected, then snap-frozen, and crushed in liquid nitrogen. Equal volume of 2 X SDS protein loading buffer was added to the powder as previously described [31]. Protein samples were subjected to electrophoresis in 12% SDS-polyacrylamide gel (SDS-PAGE) at voltage 120 V, then all proteins on the gel were transferred to a piece of nitrocellulose membrane (Cat. ab133412, Abcom, Shanghai, China). The self-prepared polyclonal anti-CP^YoMV^ was used as first antibody. Then, alkaline phosphatase (AP)-conjugated immunoglobulin (AP-A) (Sangon Biotech Co., Ltd., Shanghai, China) was utilized as a secondary antibody.

### 4.8. Sequence Analysis

All complete genomic sequences of YoMV deposited in the NCBI database were downloaded. The sequences corresponding to the whole genome, coding sequences for replicase, movement protein, and coat protein, were comparatively analyzed by phylogenetic tree construction. Here, MEGA7 software was used as previously described [50].

## Figures and Tables

**Figure 1 ijms-25-01620-f001:**
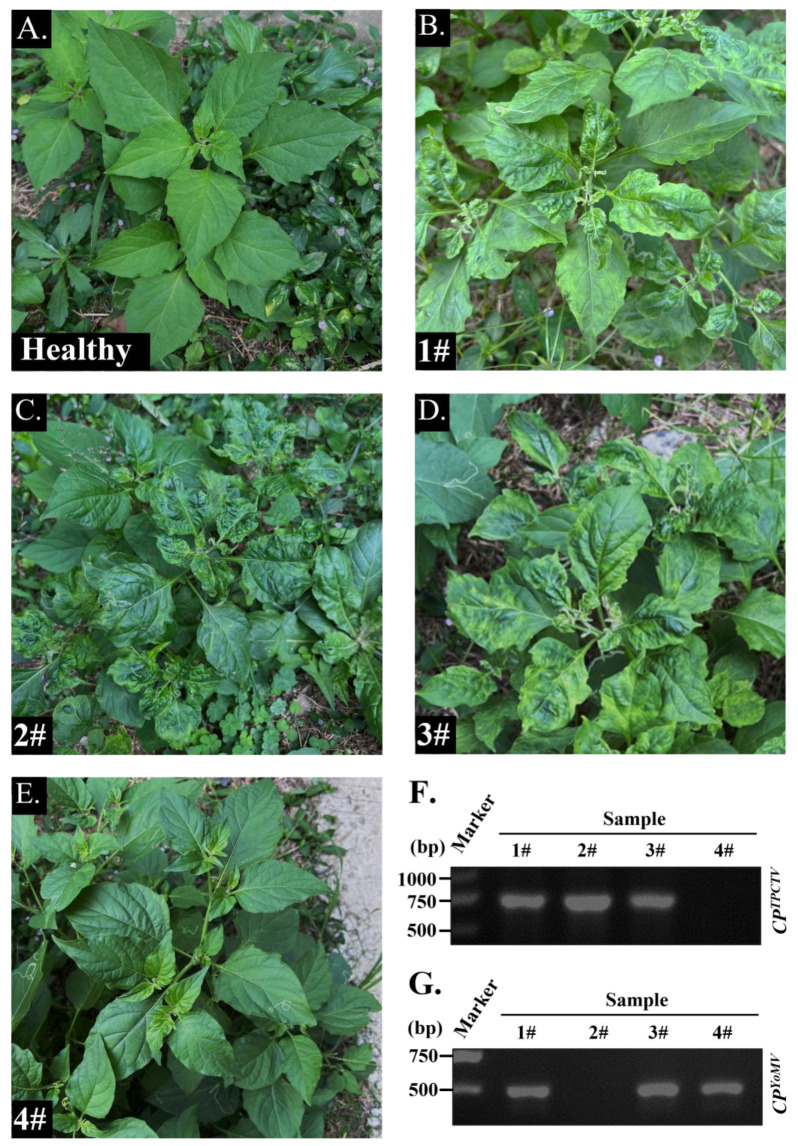
Symptoms appearing on *S. nigrum* in the campus of Yangzhou University and molecular detections. (**A**). Healthy *S. nigrum.* (**B**). *S. nigrum* sample 1#. (**C**). *S. nigrum* sample 2#. (**D**). *S. nigrum* sample 3#. (**E**). *S. nigrum* sample 4#. (**F**). RT-PCR amplification of the coat protein gene (*CP*) of youcai mosaic virus (YoMV). (**G**). RT-PCR amplification of the *CP* gene of tomato pseudo-curly top virus (TPCTV).

**Figure 2 ijms-25-01620-f002:**
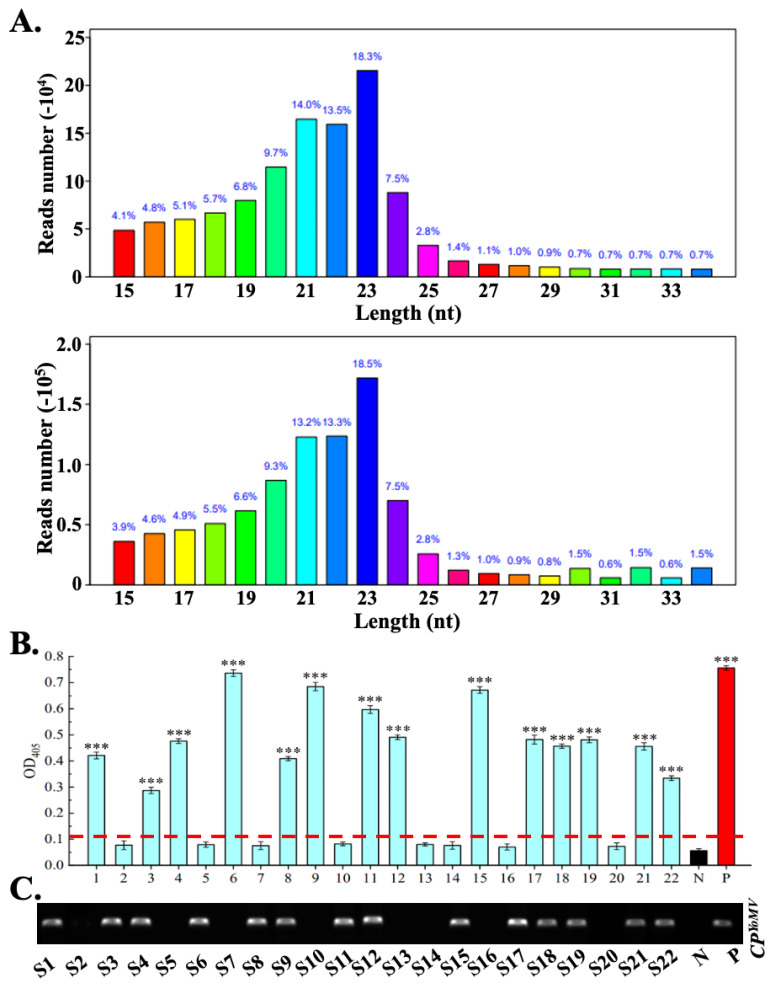
Detection of YoMV using ELISA and RT-PCR analyses. (**A**). The length distributions of all reads (upper panel) and unique reads (bottom panel) obtained from high-throughput siRNA deep-sequencing analyses. Columns with different color represent the reads with different length (range from 15 nt to 34 nt). (**B**). Histogram illustration of the ELISA results. The specific polyclonal anti-CP^YoMV^ antibodies were used in analyses. Red line indicates the positivity threshold and the OD_405_ value. Student’s *t*-tests were performed. “***” means *p*-value < 0.001. (**C**). RT-PCR detect the YoMV using the same samples tested in ELISA analyses. In RT-PCR analyses, the positive control is the amplification product that uses pCB301-YoMV-GD plasmid as a template. “P” positive control, “N” negative control.

**Figure 3 ijms-25-01620-f003:**
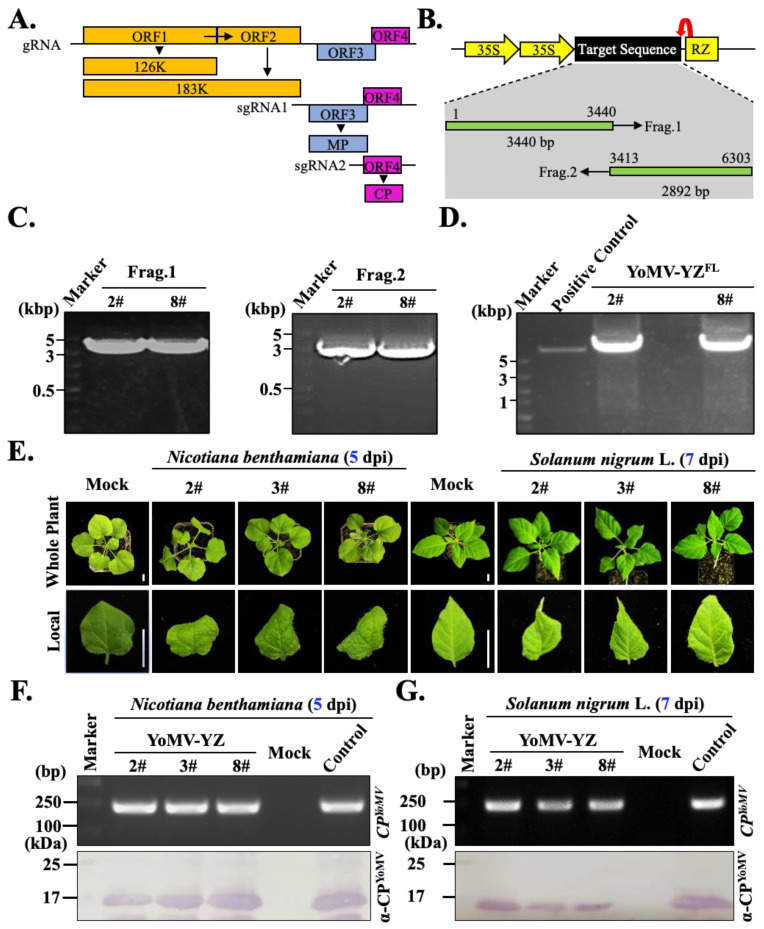
Construction of the *Agrobacterium*-infiltration-based YoMV infectious cDNA clones. (**A**). Schematic representation of the YoMV genome organization. Red arrow represents the splicing site of the ribozyme (RZ). (**B**). Schematic diagram of the construction strategy used in this study. (**C**). PCR amplification of cDNA fragments corresponding to frag1 and frag2 on the YoMV genome; FL superscript indicates the full length of YoMV genome. (**D**). Validation of the constructed full-length YoMV cDNA using PCR amplification. (**E**). Symptoms on *N. benthamiana* and *S. nigrum* inoculated with YoMV cDNA infectious clones at 5 days post-infiltration (dpi) (left panel) and 7 dpi (right panel), respectively. Plants infiltrated with *Agrobacterium* transformed with empty vectors were used as the mock treatment. Clones 2#, 3#, and 8# represent three independent infecting clones of YoMV. Bar, 1 cm. (**F**). RT-PCR and Western blot detection of YoMV on *N. benthamiana*. (**G**). RT-PCR and Western blot detection of YoMV on *S. nigrum*.

**Figure 4 ijms-25-01620-f004:**
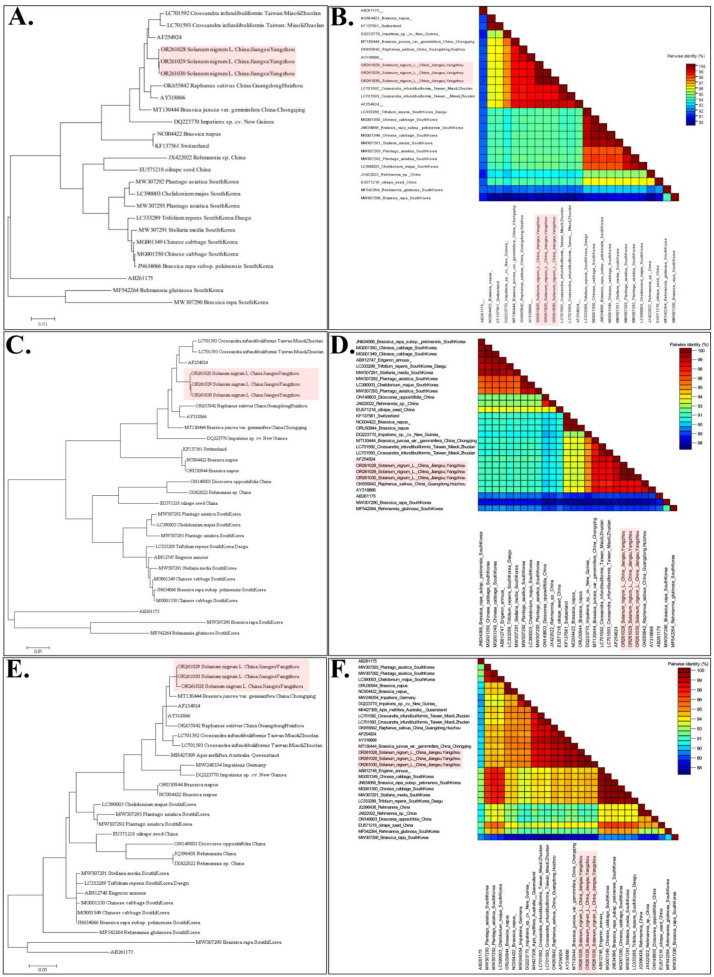
Genomic sequences and pairwise identity analyses of YoMV. (**A**). Phylogenetic tree based on genome sequences of YoMV. Bar, 0.01. (**B**). Consistency analysis of genome sequences with pairwise identify analyses. (**C**). Phylogenetic tree based on the RdRp coding sequences of YoMV. Bar, 0.01. (**D**). Consistency analysis of RdRp coding sequences. (**E**). Phylogenetic tree based on the movement protein (MP) coding sequences of YoMV. Bar, 0.01. (**F**). Consistency analysis of MP coding sequences. Red boxes represent the sequences of the YoMV-YZ isolates.

## Data Availability

Appendix A are provided in the Appendix A.

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
