# Peer review of "Molecular Characterization and Pathogenicity of an Infectious cDNA Clone of Youcai Mosaic Virus on Solanum nigrum"

_ijms, 2024, doi:10.3390/ijms25031620_

Round 1

Reviewer 1 Report

Comments and Suggestions for Authors

In the manuscript „Molecular characterisation and pathogenicity of an infectious clone of youcai mosaic virus on Solanum nigrum” by Gu et al. the authors described a new isolate of YoMV-YZ infecting diseased S.nigrum collected in the campus of Yangzhou University in Jiangsu Province, China. The identification of the virus in symptomatic plants was done utilizing next-generation sequencing of siRNA libraries prepared from total RNA extracted from the mentioned host. The siRNA-Seq reads were then taken for assembling contigs – among them, the YoMV coding sequence was identified. The presence of the virus in the plants was verified by RT-PCR with YoMV-specific primers followed by corresponding ELISA assay. Next, the sequence of YoMV-YZ was RT-PCR-amplified (in two pieces), recombined with pCA4Y plasmid (in yeast) and the resulting plasmid (pCA4Y-YoMV-YZ) was used to transform A. tumefaciens followed by test plants agroinfiltration. The progeny virus was identified (in the infiltrated plants) by RT-PCR and western blot. Additionally, the authors performed a phylogenetic analysis of the sequenced YoMV-YZ clone (full sequence, RdRP, MP and CP).

The presented results bring interesting information on YoMV biology – here the S. nigrum was shown to be the host of the virus. The phylogenetic analysis of the YoMV-YZ showed its position among other known YoMVs. The prepared infectious clone of YoMV-YZ can be used as a valuable tool to analyze virus pathogenicity, especially if it differs between isolates on different hosts. The presented results are supported variety of diagnostics assays and analyses, which makes the work notably worth publishing. However, some issues need to be clarified:

1.      How the full genomic sequence of the YoMV-YZ was assessed? In my opinion, NGS data of RNAseq will not bring sufficient information about 5` and 3` termini of virus RNA. For this 5`/3` RACE analysis needs to be done before stating that the entire gRNA of the YoMV-YZ was characterized.

2.      The main stress of the work was put on the identification of the YoMV-YZ in S. nigrum plants, whereas information on preparing the infectious clones of the virus seems to be rather poor. Before reading the entire manuscript I expected that the methodology focusing on making the clone would be described in detail – it is not and needs to be improved.

3.      As the authors indicated, the symptoms of A. tumefaciens-delivered YoMV-YZ on S. nigrum were milder than those observed on original specimens infected with the virus. These discrepancies were to be caused by VoMV/TPCTV co-infecting the S. nigrum (what was discussed by the authors). However, in plant specimen no 4 (#4) only YoMV was detected – how the disease symptoms in the plants look like? Please, include the data. Moreover, if the symptoms of agro-delivered YoMV-YZ differ from the expected, this might have arisen from mutations accumulating in its genome. The data should be indicated in the work.

Other comments:

Abstract: use the proper style for the full name of YoMV;

lines 18-19, this needs to be rewritten because it is hard to understand;

Lines 20-21: change the repeated words;

Line 23: is the ‘respectively’ used in the correct sense?

Lines 24-27: need to be written again more clearly.

Introduction: line 32: is the name of the genus written in the correct style?

Line 34: change the ‘folk remedy’.

Line 43: change the word ‘internationally’ into the correct one.

Line 56: ‘in vitro’ (as well as other Latin words) should be written in italics.

Materials and methods: please, rewrite the titles of 2.1, 2.4, 2.5 paragraphs.

Line: 80-81: is it really important where the RT kit was purchased from? I would be more interested in information on how the cDNA was prepared.

Moreover, as the ELISA was used for testing, it needs to be described how it was done -this needs to be added to the section.

Line 115-117: the information should be included in the Figures captions.

Results: line 119: is the ‘anomalous’ good word here? Please describe in more detail the symptoms observed in diseased plants.

Section 3.2: please describe the results of siRNA-seq in detail, including the data delivered from the Virus Detect application.

Line 134: what do the authors mean by writing ‘After removing of the junctions’?

Line 150-151: rewrite the sentence describing the ELISA results (it can not be stated that the ‘Samples exhibited strong colour reaction’.

Line 153: why the protein extract, but not the plants with verified YoMV infection, was used as a positive control in ELISA? The two sources of antigen must not be mixed in the same assay.

The data from 5`/3` RACE must be included in the manuscript, otherwise, it can not be stated that the full genomic sequence of the YoMV-YZ was described.

Line 190: in the figure caption the authors described results referring to clones #2, #5 and #8, whereas the figure showed some different data. This needs to be checked again.

Figure 3, panel C: what is the ‘Fl’ upper index next to the YoMV-YZ?

Lines 199-207: the information on results from Fig3, panel F is missing.

Line 209: change the ‘assembly of YoMV In nature’.

Line 213: the sentence needs to be changed.

Line 228-230: the word ‘sequence’ was used 4 times. This needs to be changed.

Lines 259, and 260: change the ‘agrobacterium-mediated’ into the correct form.

Lines 266-268: this needs to be rewritten.

Line 272: is the word ‘for’ used correctly?

Lines 253-255 could be included at the beginning of the Discussion.

Line 280: change the word ‘worsened’.

Lines 293-298: this should be rewritten.

Additionally, I would recommend the authors use the English editing service: some sentences do not make any sense right now. 

Author Response

Response to Reviewers

Dear Editors and Reviewers:

Thank you for your letter and for the reviewers’ comments concerning our manuscript entitled “Molecular Characterization and Pathogenicity of an Infectious cDNA Clone of Youcai Mosaic Virus on Solanum Nigrum” (ID: IJMS-2694701). These comments are all valuable and very helpful for revising and improving our paper, as well as the important guiding significance to our researches. We have studied comments carefully and have made correction which we hope meet with approval. The main corrections in the paper and the responds to the reviewer’s comments are as following:

Response to Reviewer 1’s Comments

In the manuscript „Molecular characterization and pathogenicity of an infectious clone of youcai mosaic virus on Solanum nigrum” by Gu et al. the authors described a new isolate of YoMV-YZ infecting diseased S.nigrum collected in the campus of Yangzhou University in Jiangsu Province, China. The identification of the virus in symptomatic plants was done utilizing next-generation sequencing of siRNA libraries prepared from total RNA extracted from the mentioned host. The siRNA-Seq reads were then taken for assembling contigs – among them, the YoMV coding sequence was identified. The presence of the virus in the plants was verified by RT-PCR with YoMV-specific primers followed by corresponding ELISA assay. Next, the sequence of YoMV-YZ was RT-PCR-amplified (in two pieces), recombined with pCA4Y plasmid (in yeast) and the resulting plasmid (pCA4Y-YoMV-YZ) was used to transform A. tumefaciens followed by test plants agroinfiltration. The progeny virus was identified (in the infiltrated plants) by RT-PCR and western blot. Additionally, the authors performed a phylogenetic analysis of the sequenced YoMV-YZ clone (full sequence, RdRP, MP and CP).

The presented results bring interesting information on YoMV biology – here the S. nigrum was shown to be the host of the virus. The phylogenetic analysis of the YoMV-YZ showed its position among other known YoMVs. The prepared infectious clone of YoMV-YZ can be used as a valuable tool to analyze virus pathogenicity, especially if it differs between isolates on different hosts. The presented results are supported variety of diagnostics assays and analyses, which makes the work notably worth publishing. However, some issues need to be clarified:

  1. How the full genomic sequence of the YoMV-YZ was assessed? In my opinion, NGS data of RNAseq will not bring sufficient information about 5` and 3` termini of virus RNA. For this 5`/3` RACE analysis needs to be done before stating that the entire gRNA of the YoMV-YZ was characterized. 

Response: We sincerely appreciate the valuable comments. Actually, we conducted a comparative analysis of the whole YoMV genome in mainland China before the gene cloning work, and found that YoMV has more conserved 5' and 3' sequences in China, based on this discovery, we designed the primers for the head and tail of the YoMV genome (as follows). the RNAseq data were only used to design primers for the middle of the genome, and we have attached the Supplementary Material Table S2, where you can view all the primer information, We have also described the primer information in detail in the text in lines 234-248.

  1. The main stress of the work was put on the identification of the YoMV-YZ in S. nigrum plants, whereas information on preparing the infectious clones of the virus seems to be rather poor. Before reading the entire manuscript I expected that the methodology focusing on making the clone would be described in detail – it is not and needs to be improved.

Response: Thank you for your valuable comments. We have added a detailed description of the experimental design and manipulation in section 3.3 (lines 231-259), while refining the information about the yeast system in Materials and Methods 2.5.

  1. As the authors indicated, the symptoms of A. tumefaciens-delivered YoMV-YZ on S. nigrum were milder than those observed on original specimens infected with the virus. These discrepancies were to be caused by VoMV/TPCTV co-infecting the S. nigrum (what was discussed by the authors). However, in plant specimen no 4 (#4) only YoMV was detected – how the disease symptoms in the plants look like? Please, include the data. Moreover, if the symptoms of agro-delivered YoMV-YZ differ from the expected, this might have arisen from mutations accumulating in its genome. The data should be indicated in the work.

Response: Thank you for your constructive comments. We apologize for the negligence of our previous work, and with your valuable suggestions, we have rechecked and corrected the relevant symptom information, the symptom pictures of all samples have been added to Fig. 1, and we have corrected the misdescriptions in the discussion (lines 343-354).

Other comments:

  1. Abstract: use the proper style for the full name of YoMV;

Response: Thank you for your professional comments. We have checked and corrected (line 19).

  1. lines 18-19, this needs to be rewritten because it is hard to understand;

Response: Thank you for your professional comments. We have changed the description to “Exploring the virus genomic evolution and constructing virus infectious cDNA clones are essential to achieve a deeper understanding of the interaction between plant host and virus.” In lines 16-18.

  1. Lines 20-21: change the repeated words;

Response: Thank you for your professional comments. We have changed in lines 21-22.

  1. Line 23: is the ‘respectively’ used in the correct sense?

Response: Thank you for your professional comments. We have deleted‘respectively’ in line 22.

  1. Lines 24-27: need to be written again more clearly.

Response: Thank you for your professional comments. We have rewritten to “We constructed the cDNA infectious clone of YoMV from Solanum nigrum L. and tested its infectivity on Nicotiana benthamiana and Solanum nigrum plants. Based on the cloned YoMV sequences, we analyzed the evolutionary relationship with existing sequences deposited in the NCBI database and clearly identified the origin of the YoMV-YZ isolate. Furthermore, the cloned YoMV provides the fundamentals for future in-depth study of YoMV.”.

  1. Introduction: line 32: is the name of the genus written in the correct style?

Response: Thank you for your professional comments. We have checked and corrected in line 32.

  1. Line 34: change the ‘folk remedy’.

Response: Thank you for your valuable comments. We have changed 'folk remedy ' to 'treatments' in line 35.

  1. Line 43: change the word ‘internationally’ into the correct one.

Response: Thank you for your valuable comments. We have corrected “internationally” in line 38.

  1. Line 56: ‘in vitro’ (as well as other Latin words) should be written in italics.

Response: Thank you for your professional comments. We have checked and corrected all of Latin words.

  1. Materials and methods: please, rewrite the titles of 2.1, 2.4, 2.5 paragraphs.

Response: Thank you for your professional comments. We have changed the title of 2.1 to “Cultivation of test plants” in line 85. We have changed the titles of 2.4 to “Construction of cDNA infectious clones of YoMV-YZ” in line 120. We have changed the titles of 2.5 to “Detection of infectious clones using Agrobacterium-mediated inoculation” in line 125.

  1. Line: 80-81: is it really important where the RT kit was purchased from? I would be more interested in information on how the cDNA was prepared.

Response: Thank you for your valuable comments. We have added a description of this in section 2.2 (lines 89-96), and we have also detailed the procedure in section 3.3 (lines 231-259) of the results.

  1. Moreover, as the ELISA was used for testing, it needs to be described how it was done -this needs to be added to the section.

Response: Thank you for your valuable comments. We have added in lines 105-119.

  1. Line 115-117: the information should be included in the Figures captions.

Response: Thank you for your valuable comments. We have moved this part of the information to the Figures captions (lines 227-228).

  1. Results: line 119: is the ‘anomalous’ good word here? Please describe in more detail the symptoms observed in diseased plants. 

Response: Thank you for your valuable comments. We have removed this inappropriate statement and added a new description of the symptoms in line 158.

  1. Section 3.2: please describe the results of siRNA-seq in detail, including the data delivered from the Virus Detect application.

Response: Thank you for your professional comments. We have described the results of siRNA-seq in detail, and relevant data were added, additional descriptions of post-sequencing data processing and analysis have also been provided in lines 175-185.

  1. Line 134: what do the authors mean by writing ‘After removing of the junctions’?

Response: Thank you for your valuable comments. we actually trying to describe adapter, we have changed the description of the error and explained the adapter (line 180).

  1. Line 150-151: rewrite the sentence describing the ELISA results (it can not be stated that the ‘Samples exhibited strong colour reaction’.

Response: Thank you for your valuable comments. We have described the ELISA results: “showed intensive yellow in ELISA detection” (line 197).

  1. Line 153: why the protein extract, but not the plants with verified YoMV infection, was used as a positive control in ELISA? The two sources of antigen must not be mixed in the same assay.

Response: Thank you for your valuable comments. In fact, the positive samples we used were exactly the plants that had been infected by YoMV (Agrobacterium-infiltrated N. benthamiana, infectious cDNA clone plasmid used was pCB301-YoMV-GD), and we likewise added to this in the manuscript in line 229-230.

  1. The data from 5`/3` RACE must be included in the manuscript, otherwise, it can not be stated that the full genomic sequence of the YoMV-YZ was described.

Response: We sincerely appreciate the valuable comments. Actually, we conducted a comparative analysis of the whole YoMV genome in mainland China before the gene cloning work, and found that YoMV has more conserved 5' and 3' sequences in China, based on this discovery, we designed the primers for the head and tail of the YoMV genome (as follows). Subsequent genome amplification was also performed based on this information

  1. Line 190: in the figure caption the authors described results referring to clones #2, #5 and #8, whereas the figure showed some different data. This needs to be checked again.

Response: Thank you for your professional comments. We have checked and corrected in line 268.

  1. Figure 3, panel C: what is the ‘Fl’ upper index next to the YoMV-YZ?

Response: Thank you for your valuable comments. 'FL' means full-length, we have added a description in the figure caption in line 264.

  1. Lines 199-207: the information on results from Fig3, panel F is missing.

Response: Thank you for your professional comments. We have checked and added in line 276.

  1. Line 209: change the ‘assembly of YoMV In nature’.

Response: Thank you for your valuable comments. We have changed ‘assembly’ to ‘recombination’ in line 283.

  1. Line 213: the sentence needs to be changed.

Response: We sincerely appreciate the valuable comments. We have changed to “First, all 22 full-length YoMV genome sequences, together with our sequences, were submitted for analyses, and a phylogenetic tree based on the genome sequences was constructed” in lines 287-289.

  1. Line 228-230: the word ‘sequence’ was used 4 times. This needs to be changed.

Response: Thank you for your valuable comments. We have rewritten to “The results showed that with respect to RdRp, YoMV-YZ showed the highest concordance with YoMV-SH” in lines 303-304.

  1. Lines 259, and 260: change the ‘agrobacterium-mediated’ into the correct form.

Response: Thank you for your professional comments. We have checked and corrected in lines 333, and 334.

  1. Lines 266-268: this needs to be rewritten.

Response: Thank you for your professional comments. We have rewritten to “The Agrobacterium-mediated YoMV-YZ infectious cDNA clone is a valuable tool, and the theory and materials in our study lay the foundation for further comparative studies in virology.” in lines 343-345.

  1. Line 272: is the word ‘for’ used correctly?

Response: Thank you for your valuable comments. We have changed ‘for’ to ‘from’.

  1. Lines 253-255 could be included at the beginning of the Discussion.

Response: We sincerely appreciate the valuable comments. We have moved Lines 253-255 to the beginning of the discussion.

  1. Line 280: change the word ‘worsened’.

Response: Thank you for your professional comments. We've rewritten this sentence to replace the inappropriate description in it in lines 358.

  1. Lines 293-298: this should be rewritten.

Response: Thank you for your professional comment. We have rewritten to “Comparing the sequences of the MP/CP part with those of other strains around the world, the YZ strain is more closely related to other strains in China and most distantly related to strains in South Korea (MW307290), and the sequences of these two parts of YoMV still show a positive correlation with spatial distances around the world. Although the biological significance of these mutations or recombination in YoMV re-mains unclear, mutations and recombination in RNA viruses bear significant im-portance for generating genetic variation, adapting to new hosts, evading host immune responses, and emerging as new infectious agents [50]. However, further re-search is necessary to gain more profound insights.” in lines 380-388.

  1. Additionally, I would recommend the authors use the English editing service: some sentences do not make any sense right now.

Response: Thank you for your professional comment, we have optimized the English presentation as much as possible and conducted English editing services for the manuscript in order to improve the manuscript.

Reviewer 2 Report

Comments and Suggestions for Authors

In their manuscript "Molecular characterization and pathogenicity of an infectious cDNA clone of youcai mosaic virus on Solanum nigrum" Gu and co-workers collected 4 samples from symptomatic nightshade plants on the campus of the Yangzhou University. They extracted RNAs from these plants and constructed small RNA libraries. After sequencing they retrieved sequences from two viruses: YoMV and TPCTV. They focused their work on the tobamovirus YoMV and constructed three cDNA infectious agro-infiltrable clones. The three clones were indeed infectious on both the model plant N. benthamiana and S. nigrum. They used the complete sequence of the three clones and 22 full-length sequences from GenBank to construct a phylogenetic tree, that allowed them to conclude that their three clones were clustering together and were the closest to YoMV-SH and YoMV-Taiwan isolates.

They also collected 18 additional samples from the campus and 28 from 5 supplementary cities in the province.

Although it seems that an infectious clone was already available for a different isolate of this virus, this work describing an isolate of YoMV in a new host is interesting. I however have a few concerns that should be addressed before publishing.

There is some redundancy in the abstract, which should be thoroughly edited for better clarity. Some material and method issues should be clarified (see line per line comments).

Figure 2: It is not clear how the Student's t-test was used on ELISA values of single samples as this statistical test should only be performed on populations with normal distribution.

It is not clear whether there is a relation between the collected samples positive for YoMV (1#, 3# and 4# in figure 1C) and the three obtained infectious clones.

Also, the infection rate reached with the infectious clones should be indicated. How many plants were infected and how many were inoculated?

Fig. 4 is too small, one cannot see the claimed results.

The authors claim that the work has expanded our understanding of the virus' host range: this work cannot be considered a study of the host range of YoMV.

in their smal "survey", they found an occurrence of YoMV reaching 80 % in Yancheng city, however it is not clear whether samples were randomly collected from symptomatic and symptomless plants or whether symptomatic plants were sampled.

The authors discuss that the milder symptoms obtained from the infectious clones compared to the original samples could be attributed to single infection whereas original samples were also infected with TPCTV. This is however not the case for sample 4 which was positive only for YoMV (negative for TPCTV). Was this sample less symptomatic than the 3 others?

Lines 32-33: no verb in the sentence

Lines 36-41: what the plant is said good for is not of great interest for the present work. This could be removed from the introduction

Line 42: viruses infect plants, they do not infest them.

Line 56: correct works.

line 58: "Brassica napus" should be in italics.

Line 61: two verbs.

Line 66: remove "target".

Line 68: remove "that".

line 69: replace "people's" with "our".

Line 78: it is not clear what "which beings to the plant virus laboratory" means

Line 85: it is not clear what the RNA was mixed with.

Lines 93-94: RNA cannot be infected.

Line 95: it is not clear how the fragments were purified. Was it on agarose gel? or using a kit?

Line 92: were several clones obtained from several starting plants?

Line 95: what strain? YPH500? Start the § with a sentence saying that you used the homologous recombination-based cloning method in yeast to produce the cDNA clone.

How was the yeast transformation performed?

§ 2-7: this paragraph should better explain the ELISA: how were the samples prepared, what antibodies/serum was used…

Line 123: how were the samples "preserved"? what were the preservation conditions?

Line 134: remove "of", remove "and".

Line 136: replace " in the analysis’s total reads" with "of the total reads".

Line 137: from "It was showed that…" shall we understand that sequences from the two viruses were retrieved from the blastn?

Lines 145-146: please clarify "siRNA high throughput sequencing was firstly showed 145 that YoMV infects S. nigrum naturally in wild field."

Fig 2 should be split because it describes very different results. Part A reports on the siRNAs extracted from 4 samples collected in September and used to retrieve the virus sequences (TPCTV and YoMV), whereas parts B to D are about 22 other collected samples, tested both by ELISA and RT-PCR for the presence of YoMV. ELISA is not considered a molecular but an immunologic assay, please correct caption. Panel B is of no interest as panel C is more informative. A line indicating the positivity threshold could be added in the histogram. Line 171: "mean" and "is" are redundant. Lines 176-177: please indicate how the primers were designed, was a specific sequence used? was a consensus sequence used? If so indicate how the consensus sequence was obtained. Or were primers from the literature used?

Line 181: how many benthamiana plants were infiltrated?

Line 185: the antibodies used in this western blot should be described in the mat & met section or a reference should be indicated.

Fig. 3: Panels C and D, what do #2 and #8 stand for? Line 196: change #5 to #3.

Line 202: leaf curling?

Line 203: what is green degradation? is it chlorosis?

Line 206: correct "systemically infect the model plant…"

Lines 216-218: It is difficult to see that "our three isolates form a self-contained cluster and are most closely related to YoMV-SH isolate and YoMV-Taiwan isolate, and most distantly related to the South African isolates" as they are not highlighted in the figure or at least their number given in the text.

Lines 22§-238: please rephrase as this is not clear.

Line 262: better explain that the infectious clones provide a tool to perform reverse genetic experiments.

References: remove extra capitals in the titles.

Comments on the Quality of English Language

The english should be improved for better clarity.

Author Response

Response to Reviewers

Dear Editors and Reviewers:

Thank you for your letter and for the reviewers’ comments concerning our manuscript entitled “Molecular Characterization and Pathogenicity of an Infectious cDNA Clone of Youcai Mosaic Virus on Solanum Nigrum” (ID: IJMS-2694701). These comments are all valuable and very helpful for revising and improving our paper, as well as the important guiding significance to our researches. We have studied comments carefully and have made correction which we hope meet with approval. The main corrections in the paper and the responds to the reviewer’s comments are as following:

Response to Reviewer 2’s Comments

In their manuscript "Molecular characterization and pathogenicity of an infectious cDNA clone of youcai mosaic virus on Solanum nigrum" Gu and co-workers collected 4 samples from symptomatic nightshade plants on the campus of the Yangzhou University. They extracted RNAs from these plants and constructed small RNA libraries. After sequencing they retrieved sequences from two viruses: YoMV and TPCTV. They focused their work on the tobamovirus YoMV and constructed three cDNA infectious agro-infiltrable clones. The three clones were indeed infectious on both the model plant N. benthamiana and S. nigrum. They used the complete sequence of the three clones and 22 full-length sequences from GenBank to construct a phylogenetic tree, that allowed them to conclude that their three clones were clustering together and were the closest to YoMV-SH and YoMV-Taiwan isolates. 

They also collected 18 additional samples from the campus and 28 from 5 supplementary cities in the province. 

Although it seems that an infectious clone was already available for a different isolate of this virus, this work describing an isolate of YoMV in a new host is interesting. I however have a few concerns that should be addressed before publishing.

  1. There is some redundancy in the abstract, which should be thoroughly edited for better clarity. Some material and method issues should be clarified (see line per line comments).

Response: Thank you for your professional comments. We have rewritten the abstract section (lines 15-28).

  1. Figure 2: It is not clear how the student’st-test was used on ELISA values of single samples as this statistical test should only be performed on populations with normal distribution.

Response: Thank you for your valuable question. I'm sorry that my presentation was incomplete and led to your misunderstanding. In fact, I conducted a T test on the OD reading of each sample separately against the negative control reading, just to show that our data is reliable.

  1. It is not clear whether there is a relation between the collected samples positive for YoMV (1#, 3# and 4# in figure 1C) and the three obtained infectious clones.

Response: Thank you for your valuable question. They are not linked, I have added a description of the template used for gene cloning in the text, it is a mixture of samples, so it is not possible to determine the relationship between them, (line 240-241).

  1. Also, the infection rate reached with the infectious clones should be indicated. How many plants were infected and how many were inoculated?

Response: Thank you for your professional comment. We have added (lines 277-279).

  1. 4 is too small; one cannot see the claimed results.

Response: Thank you for your valuable comment. We have enlarged the size of Figure 4 to make it easier to observe it more clearly.

  1. The authors claim that the work has expanded our understanding of the virus' host range: this work cannot be considered a study of the host range of YoMV.

Response: Thank you for your professional comment. We apologize that we did previously overstate the research, and we have checked and corrected it.

  1. in their small "survey", they found an occurrence of YoMV reaching 80 % in Yancheng city, however it is not clear whether samples were randomly collected from symptomatic and symptomless plants or whether symptomatic plants were sampled.

Response: Thank you for your valuable question. We have more clearly described the samples collected as having symptoms of suspected virus infection in line 207.

  1. The authors discuss that the milder symptoms obtained from the infectious clones compared to the original samples could be attributed to single infection whereas original samples were also infected with TPCTV. This is however not the case for sample 4 which was positive only for YoMV (negative for TPCTV). Was this sample less symptomatic than the 3 others?

Response: Thank you for your constructive comments. We apologize for the negligence of our previous work, and with your valuable suggestions, we have rechecked and corrected the relevant symptom information, the symptom pictures of all samples have been added to Fig. 1, and we have corrected the misdescriptions in the discussion (lines 343-354).

  1. Lines 32-33: no verb in the sentence

Response: Thank you for your professional comment. We have checked and corrected (lines 32-33).

  1. Lines 36-41: what the plant is said good for is not of great interest for the present work. This could be removed from the introduction

Response: Thank you for your professional comment. We have deleted.

  1. Line 42: viruses infect plants, they do not infest them.

Response: Thank you for your professional comment. We have checked and corrected.

  1. Line 56: correct works.

Response: Thank you for your professional comment. We have checked and reorganized the language in lines 55-56.

  1. line 58: "Brassica napus" should be in italics.

Response: Thank you for your professional comment. We have corrected .

  1. Line 61: two verbs.

Response: Thank you for your valuable comment. We have checked and corrected in line 71.

  1. Line 66: remove "target".

Response: We sincerely appreciate the valuable comment. We have removed.

  1. Line 68: remove "that".

Response: We sincerely appreciate the valuable comment. We have removed.

  1. line 69: replace "people's" with "our".

Response: We sincerely appreciate the valuable comment. We have replaced "people's" with "our". Line 79.

  1. Line 78: it is not clear what "which beings to the plant virus laboratory" means

Response: We sincerely appreciate the valuable comment. We have changed this unclear description to ‘The greenhouse is located at the plant virus laboratory of Yangzhou University’, line 87-88.

  1. Line 85: it is not clear what the RNA was mixed with.

Response: Thank you for your valuable question. We have changed this unclear description to ‘The S. nigrum samples were mixed in equal parts’, line 98.

  1. Lines 93-94: RNA cannot be infected.

Response: Thank you for your professional comment. We have changed this unclear description to ‘from RNA of S. nigrum containing YoMV’, line 122-123.

  1. Line 95: it is not clear how the fragments were purified. Was it on agarose gel? or using a kit?

Response: Thank you for your professional comment. We have added descriptions, line 123.

  1. Line 92: were several clones obtained from several starting plants?

Response: Thank you for your valuable question. They are not linked, I have added a description of the template used for gene cloning in the text, it is a mixture of samples, so it is not possible to determine the relationship between them, (line 240-241).

  1. Line 95: what strain? YPH500? Start the § with a sentence saying that you used the homologous recombination-based cloning method in yeast to produce the cDNA clone.

Response: Thank you for your valuable question. We have supplemented the strains YPH500 in line 6.

  1. How was the yeast transformation performed?

Response: Thank you for your valuable question. We have supplemented the kits we use for yeast transformations in line 125-126.

  1. 2-7: this paragraph should better explain the ELISA: how were the samples prepared, what antibodies/serum was used…

Response: Thank you for your professional comment. We have added methods and materials for ELISA detection in line 105-119.

  1. Line 123: how were the samples "preserved"? what were the preservation conditions?

Response: Thank you for your valuable question. We have added the relevant conditions ‘freeze-dried and stored in a -80°C refrigerator’ in line 160.

  1. Line 134: remove "of", remove "and".

Response: We sincerely appreciate the valuable comment. We have removed "of" "and". Line 179.

  1. Line 136: replace " in the analysis’s total reads" with "of the total reads".

Response: We sincerely appreciate the valuable comment. We have replaced " in the analysis’s total reads" with "of the total reads" in line 178.

  1. Line 137: from "It was showed that…" shall we understand that sequences from the two viruses were retrieved from the blastn?

Response: Thank you for your valuable question. We have added the description in line 178-182. You can understand that the sequence after siRNA sequencing and splicing matched these two viruses in the database

  1. Lines 145-146: please clarify "siRNA high throughput sequencing was firstly showed 145 that YoMV infects nigrum naturally in wild field."

Response: Thank you for your valuable question. We actually want to show that this is the first evidence that YoMV can infect S. nigrum. However, it is misleading here, so we have removed the description of ‘firstly’ in line190-191.

  1. Fig 2 should be split because it describes very different results. Part A reports on the siRNAs extracted from 4 samples collected in September and used to retrieve the virus sequences (TPCTV and YoMV), whereas parts B to D are about 22 other collected samples, tested both by ELISA and RT-PCR for the presence of YoMV. ELISA is not considered a molecular but an immunologic assay, please correct caption. Panel B is of no interest as panel C is more informative. A line indicating the positivity threshold could be added in the histogram. Line 171: "mean" and "is" are redundant. Lines 176-177: please indicate how the primers were designed, was a specific sequence used? was a consensus sequence used? If so indicate how the consensus sequence was obtained. Or were primers from the literature used?

Response: Thank you for your professional comments. We are sorry for your misunderstanding. Actually, the samples used in Figure 2A are samples 1-4 in Figure 2B, and samples 5-22 in Figure 2B are other samples from the campus. We believe that samples 1-22 are all from the campus, so the test results of them can be combined, but to avoid the same misunderstanding, we have explained the origin of samples S5-S22 in line 193; we have corrected the title in Figure 2 in line 221;we have removed panel B and added a positive threshold line to the histogram;we have deleted "mean" and "is" in line 229;we have described the primer design idea in detail in lines 233-247, and have listed all primer information in Supplemental materials Table 2.

  1. Line 181: how many benthamiana plants were infiltrated?

Response: Thank you for your valuable question. We added the relevant information “For each positive monoclone, we performed 10 repetitions on each plant, as expected, all of the inoculated plants were systematically infected, with the results showing an infection rate of 100%.”in line 276-278.

  1. Line 185: the antibodies used in this western blot should be described in the mat & met section or a reference should be indicated.

Response: We sincerely appreciate the valuable comment. We have supplemented the methods of the Western blot detection in line 141-150, and have supplemented the information on the antibodies used in Western blot detection in line 146-149.

  1. 3: Panels C and D, what do #2 and #8 stand for? Line 196: change #5 to #3.

Response: Thank you for your valuable question. #2 and #8 stand for infectious clones 2 and 8, and we have changed #5 to #3 in line 268.

  1. Line 202: leaf curling?

Response: Thank you for your valuable comment. We have checked and corrected in line 274.

  1. Line 203: what is green degradation? is it chlorosis?

Response: Thank you for your valuable question. I'm sorry for the misunderstanding, it is not chlorosis, we want to describe is mottled leaves, and we have corrected it in line 275.

  1. Line 206: correct "systemically infect the model plant…"

Response: Thank you for your valuable comment. We have checked and corrected in line 280.

  1. Lines 216-218: It is difficult to see that "our three isolates form a self-contained cluster and are most closely related to YoMV-SH isolate and YoMV-Taiwan isolate, and most distantly related to the South African isolates" as they are not highlighted in the figure or at least their number given in the text.

Response: Thank you for your professional comments. I apologize for our carelessness, but we have added a login number after each sequence described in this section (line 283-313), consistent with the information in Figure 4.

  1. Lines 22§-238: please rephrase as this is not clear.

Response: We sincerely appreciate the valuable comment. We have rephrased in line 300-313.

  1. Line 262: better explain that the infectious clones provide a tool to perform reverse genetic experiments.

Response: We sincerely appreciate the valuable comment. We have added an explanation of this sentence “In future studies, this tool can be used to explore the relationship between YoMV and its host plants, including, but not limited to elucidating its pathogenic mechanism.” in line 337-339.

  1. References: remove extra capitals in the titles.

Response: Thank you for your professional comment. We have removed the extra capitals in the titles from all the references.

Reviewer 3 Report

Comments and Suggestions for Authors

As far as I understand the paper reports a significant activity of retrieval of Solanum nigrum samples infected by Youcai Mosaic Virus by means of ELISA assays and RT-PCR reaction. It is uncleat the means of the note  "for the first time under natural conditions" .

Through the sequencing of some genomes, their characterization and comparison with those in genbank has been reported, without clear evidences of the differences.

It is unclear whether the ELISA assay was performed with a commercially available or self-produced kit, the role of the other virus found, and if co-infections with other viruses were searched for.

The preparation of the cDNA of three isolates, is claimed as a potential use for reverse genetic tools in further studying of the interactions between YoMV and plant. Despite there is no doubt that the results will contribute to deeper investigation of the virus biological characteristics, too much research is needed to claim for phytopathological and medical applications confusely indicated. The English presentation is not conducive to comprehension.

However, scarce information are given about the previous studies carried out in China and worldwide on other plants species and it would be useful that results available are cited and compared.

The reference list is too extensive, it says little about Youcai mosaic virus and its spread worldwide, while it includes 19 references on possible medical applications that are attributed to Solanum and others on different viruses that afflict this species. 

In addition, the compilation of the titles in the reference list is uneven and apparently does not comply with the journal's standards.

Comments on the Quality of English Language

A revision of the english will be helpful to explain the research done and the final aim.

Author Response

Response to Reviewers

Dear Editors and Reviewers:

Thank you for your letter and for the reviewers’ comments concerning our manuscript entitled “Molecular Characterization and Pathogenicity of an Infectious cDNA Clone of Youcai Mosaic Virus on Solanum Nigrum” (ID: IJMS-2694701). These comments are all valuable and very helpful for revising and improving our paper, as well as the important guiding significance to our researches. We have studied comments carefully and have made correction which we hope meet with approval. The main corrections in the paper and the responds to the reviewer’s comments are as following:

Response to of Reviewer 3’ Comments

The work develops a significant activity of retrieval of Solanum nigrum samples infected by YoMV by means of ELISA assays and, through the sequencing of some genomes, proceeds to the characterization and comparison with those in GenBank. Following the preparation of the cDNA of three of them, the possibility of use for medical uses is envisaged.

As far as I understand the paper reports a significant activity of retrieval of Solanum nigrum samples infected by Youcai Mosaic Virus by means of ELISA assays and RT-PCR reaction. It is uncleat the means of the note“for the first time under natural conditions”. 

  1. Through the sequencing of some genomes, their characterization and comparison with those in genbank has been reported, without clear evidences of the differences.

Response:

Thank you for your professional comments. I apologize for our carelessness, but we have added a login number after each sequence described in this section (line 283-313), consistent with the information in Figure 4, this helps us to present the described results to you more clearly.

  1. It is unclear whether the ELISA assay was performed with a commercially available or self-produced kit, the role of the other virus found, and if co-infections with other viruses were searched for.

Response:

Thank you for your professional comment. We have added methods and materials for ELISA detection in line 105-119. First of all, the infection of other viruses was not found in the siRNA sequencing results. Secondly, regarding YoMV and TPCTV, these two viruses are different types of viruses, and the CP protein similarity is very low(as shown in the figure below), so our homemade ELISA kit is reliable.

  1. The preparation of the cDNA of three isolates, is claimed as a potential use for reverse genetic tools in further studying of the interactions between YoMV and plant. Despite there is no doubt that the results will contribute to deeper investigation of the virus biological characteristics, too much research is needed to claim for phytopathological and medical applications confusely indicated. The English presentation is not conducive to comprehension.

Response: We sincerely appreciate the valuable comments. We have removed a lot of applied research on plant pathology and medicine, keeping the rest solely to show that our work is meaningful, we hope that these changes will help readers understand our work more clearly. In addition, we have optimized the English presentation as much as possible and conducted English editing services for the manuscript in order to improve the manuscript.

Varshney, P.; Vishwakarma, P.; Sharma, M.; Saini, M.; Bhatt, S.; Singh, G. K.; Saxena, K. K., Cardioprotective effect of Solanum nigrum against doxorubicin induced cardiotoxicity-an experimental study International journal of basic clinical pharmacology 2016, 5, 748-753.

Nitish, B.; Pratim, M. P.; Abhinit, K.; Atul, T.; Tasneem, A.; Uzzaman, K. M., Evaluation of cardio protective Activity of Methanolic Extract Of Solanum Nigrum Linn. in Rats. International Journal of Drug Development Research 2011, 3.

Joshi, V. K.; Joshi, A.; Dhiman, K. S., The Ayurvedic Pharmacopoeia of India, development and perspectives. Journal of ethnopharmacology 2017, 197, 32-38.

Harborne, J. B., Indian Medicinal Plants. A Compendium of 500 Species. Vol.1; Edited by P. K. Warrier, V. P. K. Nambiar and C. Ramankutty. Journal of Pharmacy and Pharmacology 2011, 46, (11), 935-935.

Mogla, E. H. O.; Abdalla, O. M.; Koko, W. S.; Saadabi, A. M., In vitro Anticancer Activity and Cytotoxicity of Solanum nigrum on Cancers and Normal Cell Lines. International Journal of Cancer Research 2014,10, 74-80.

Lai, Y. J.; Tai, C. J.; Wang, C. W.; Choong, C. Y.; Lee, B. H.; Shi, Y. C.; Tai, C. J., Anti-Cancer Activity of Solanum nigrum (AESN) through Suppression of Mitochondrial Function and Epithelial-Mesenchymal Transition (EMT) in Breast Cancer Cells. Molecules 2016, 21, (5).

Nawaz, A.; Jamal, A.; Arif, A.; Parveen, Z., In vitro cytotoxic potential of Solanum nigrum against human cancer cell lines. Saudi Journal of Biological Sciences 2021, 28, (8), 4786-4792.

Di Lorenzo, C.; Colombo, F.; Biella, S.; Stockley, C.; Restani, P., Polyphenols and Human Health: The Role of Bioavailability. Nutrients 2021, 13, (1).

Javed, T.; Ashfaq, U. A.; Riaz, S.; Rehman, S.; Riazuddin, S., In-vitro antiviral activity of Solanum nigrum against Hepatitis C Virus. Virology journal 2011, 8, 26.

Sharma, D.; Joshi, M.; Apparsundaram, S.; Goyal, R. K.; Patel, B.; Dhobi, M., Solanum nigrum L. in COVID-19 and post-COVID complications: a propitious candidate. Mol Cell Biochem 2023, 1-20.

Wang, Y.; Xiang, L.; Yi, X.; He, X., Potential Anti-inflammatory Steroidal Saponins from the Berries of Solanum nigrum L. (European Black Nightshade). J Agric Food Chem 2017, 65, (21), 4262-4272.

Kuete, V., Physical, Hematological, and Histopathological Signs of Toxicity Induced by African Medicinal Plants. In Toxicological Survey of African Medicinal Plants, 2014; pp 635-657.

Tull, D.; Earney, M.; Larke, J.; Teague, J.; Rippe, S.; Miller, G. O., Edible and Useful Wild Plants of the Southwest. In Edible and Useful Plants of the Southwest, University of Texas Press: 2013; pp 7-143.

  1. However, it does not state that Youcai mosaic virus has been detected and studied (also in China) on other plants species (CABI Compendium) and it would be useful for them to be cited and compared.

Response: Thank you for your valuable comment. We have added information on the infection of other plants by YoMV in China (line 74-77) and references (below) in the hope that this will help you learn more about the host of YoMV.

Dong, J. L.; Li, Y.; Ding, W. L.; Wang, R.,. First report of broad bean wilt virus 2 and youcai mosaic virus infecting woolly foxglove (Digitalis lanata). Journal of Plant Pathology 2017, 99, (3), 816.

Zhang, S. B.; Liu, J.; Zhao, Z. B.; Zheng, L. M.; Zhang, D. Y.; Liu, Y.; Du, J.; Peng, J.; Yan, F.; Li, F.; Xie, Y.; Cheng, Z. B., First report of pepper (Capsicum annuum) as a natural host plant for youcai mosaic virus. Plant Disease 2016, 100, (2), 541-542.

Park, C. Y.; Lee, M. A.; Lee, S. H.; Kim, J. S.; Kim, H. G. First report of youcai mosaic virus in daisy fleabane (Erigeron annuus). Plant Disease 2016, 100, (6), 1250.

Qin Y.; Wang F.; Cai L.; Gao S.; Wen Y.; Liu Y.; Lu C.; Yang J.; Li X.; Qi W.; Zhang H.; Wang F. First report of youcai mosaic virus infecting yam in china. Plant Disease 2022. 22, (5), 1026.

  1. The reference list is too extensive, it says little about Youcai mosaic virus and its spread worldwide, while it includes 19 references on possible medical applications that are attributed to Solanum and others on different viruses that afflict this species. 

Response: We sincerely appreciate the valuable comments. We removed most of the literature of uninterest, including the use of Solanum in medicine, and only kept the references to viruses infecting the plant because we added content other than those viruses.

  1. In addition, the compilation of the titles in the reference list is uneven and apparently does not comply with the journal's standards.

Response: Thank you for your professional comment. We have removed the extra capitals in the titles from all the references. Moreover, we have further checked and corrected the format.

Round 2

Reviewer 1 Report

Comments and Suggestions for Authors

Dear Authors,

thank you for resubmitting your work. I can see that the manuscript was improved in the way I asked or suggested to do. However, there are still lots of issues that need to be changed. First, I would recommend using the English editing service to improve the manuscript in this field - I believe that extensive correction of the used English will be beneficial for the understanding of the work. Sometimes less is still enough and will cover all the information you want to describe to readers. After checking the English I would advise you to resubmit the manuscript once again. This will help reviewers dive into the scientific layer of the work.

Regarding the sequence analysis of the YoMV at the 5` and 3` ends- what does it mean, that the sequence is "relatively conserved"? It suggests that it is not the same for all YoMV isolates hence designed primers might not be the best for amplification of the entire virus genome. Please include the comparative analysis of 5`/3` ends of YoMV in the supplementary data file.

Comments on the Quality of English Language

I would recommend using an English editing service before further submission. Some phrases are still hard to understand, and sentences have many repetitious words. Some words were chosen incorrectly in the wrong context. 

Author Response

Response to Reviewers

Dear Editors and Reviewers:

Thank you again for your letter and for the reviewers’ comments concerning our manuscript entitled “Molecular Characterization and Pathogenicity of an Infectious cDNA Clone of Youcai Mosaic Virus on Solanum Nigrum” (ID: IJMS-2694701). These comments are all valuable and very helpful for revising and improving our paper, as well as the important guiding significance to our researches. We have studied comments carefully and have made correction which we hope meet with approval. The main corrections in the paper and the responds to the reviewer’s comments are as following:

Comments of Reviewer 3

Comments and Suggestions for Authors

Dear Authors,

thank you for resubmitting your work. I can see that the manuscript was improved in the way I asked or suggested to do. However, there are still lots of issues that need to be changed. First, I would recommend using the English editing service to improve the manuscript in this field - I believe that extensive correction of the used English will be beneficial for the understanding of the work. Sometimes less is still enough and will cover all the information you want to describe to readers. After checking the English I would advise you to resubmit the manuscript once again. This will help reviewers dive into the scientific layer of the work.

Response:

We sincerely accept your valuable comments about English writing. We have rewritten all descriptions from beginning to end using more scientific language, and have used English editing services to make our manuscript clearer and more rigorous, hopefully in line with journal requirements.

Regarding the sequence analysis of the YoMV at the 5` and 3` ends- what does it mean, that the sequence is "relatively conserved"? It suggests that it is not the same for all YoMV isolates hence designed primers might not be the best for amplification of the entire virus genome. Please include the comparative analysis of 5`/3` ends of YoMV in the supplementary data file.

Response:

Thank you for your professional comment. I am sorry for the unclear description before, we have further added all the details of this work process in lines 226-235, and put the analysis results into the Supplementary Material Figure S3, we hope that these results can further improve the scientific nature of our manuscript.

Comments on the Quality of English Language

I would recommend using an English editing service before further submission. Some phrases are still hard to understand, and sentences have many repetitious words. Some words were chosen incorrectly in the wrong context.

Response:

We sincerely accept your valuable comments about English writing. We have rephrased difficult phrases and reduced the number of repetitive words, we also used English editing service to check and correct the grammatical errors in the text, hoping that our manuscript would be more reasonable.

Reviewer 3 Report

Comments and Suggestions for Authors

The paper has been improved and is more clear. Few minor revisions concern:

-line 32, delete belonging to the genus Solanum ....since it is already included in the species name;

-over all, delete L. for Linneum ....S. nigrum is more correct;

- two references at the end of the list are without number ...it is hard to understand if are included or not in the paper or must be deleted

Author Response

Response to Reviewers

Dear Editors and Reviewers:

Thank you again for your letter and for the reviewers’ comments concerning our manuscript entitled “Molecular Characterization and Pathogenicity of an Infectious cDNA Clone of Youcai Mosaic Virus on Solanum Nigrum” (ID: IJMS-2694701). These comments are all valuable and very helpful for revising and improving our paper, as well as the important guiding significance to our researches. We have studied comments carefully and have made correction which we hope meet with approval. The main corrections in the paper and the responds to the reviewer’s comments are as following:

Comments of Reviewer 3

Comments and Suggestions for Authors

The paper has been improved and is more clear. Few minor revisions concern:

-line 32, delete belonging to the genus Solanum ....since it is already included in the species name;

Response:

Thank you for your professional comment. We have deleted (line 32).

-over all, delete L. for Linneum ....S. nigrum is more correct;

Response:

We sincerely accept your valuable comments. We have checked and deleted all “L.” except for the first introduction to this plant.

- two references at the end of the list are without number ...it is hard to understand if are included or not in the paper or must be deleted

Response:

Thank you for your valuable comment. We have deleted the excess references.

Round 3

Reviewer 1 Report

Comments and Suggestions for Authors

Dear Authors,

 because this is the third time I was asked to review the manuscript I decided to reject it for further publication. The work was improved as it was suggested earlier. However, the sound of English in the work is still poor. My job was to check the scientific quality of the work - it is average. But the quality of the language needs further check.

There are some issues (taken only from Material and Methods) that need to be changed:

(148-149) Data obtained from ELISA reader were treated by Excel software (Version 16.76.1), and the average reading was calculated – rewrite the sentence.

(153-158) For constructing of cDNA infectious clone of YoMV-YZ isolates, two overlapping fragments of YoMV, including Frag1 (3440 bp) and Frag2 (2892 bp) were amplified using the specific pairs of primers from prepared cDNA from YoMV-infected S. nigrum. Fragments 1 and 2 products were purified using Agarose Gel DNA Extraction Kit (Cat. 11696505001, Roche, Shanghai, China) – rewrite the sentence.

(164-165) Agrobacterium- infiltration-based inoculation of the YoMV-YZ  CHANGE TO: Agrobacterium tumefaciens-mediated plant inoculation (agroinfiltration)

(169-170) Agrobacterium tumefaciens that transformed with pCA4Y empty plasmid was used as a negative control. CHANGE TO: A. tumefaciens transformed with pCA4Y plasmid were used as a negative control.

(183-185) Protein samples were subjected to electrophoresis in 12% SDS-polyacrylamide gel (SDS-PAGE) at voltage at 120 V, then all proteins on the gel were transferred to a piece of nitrocellulose membrane (Cat. ab133412, Abcom, Shanghai, China). CHANGE TO: The soluble proteins were fractioned employing SDS-PAGE and transferred to a nitrocellulose filter.

(186-188) The self-prepared polyclonal anti-CPYoMV was used as first antibody. CHANGE TO The membrane was incubated with a primary antibody (targeting YoMV coat protein) diluted X-times in a blocking solution.

 (188-190) Then, alkaline phosphatase (AP)-conjugated immunoglobulin (AP-A) (Sangon Biotech, CA, USA) was utilized as a secondary antibody. CHANGE TO Next, after several washes the membrane was incubated with a secondary antibody (conjugated with alkaline phosphatase) diluted X-times in a blocking solution.

(192-193) All complete genomic sequences of YoMV deposited in the NCBI database were download. CHANGE TO: Complete genomic sequences of YoMV were downloaded from the Genbank database. Next comparative sequence analysis was prepared considering: the full YoMV genomes, RDRP, MP, or CP coding regions.

In line 63 the Authors indicated the the YoMV encodes for two replicase subunits: 125 kDa and 182 kDa. Which one was taken for comparative analyses (line 194)? Please specify.

Comments on the Quality of English Language

English is very difficult to understand/incomprehensible.